# Reshaping of Italian Echocardiographic Laboratories Activities during the Second Wave of COVID-19 Pandemic and Expectations for the Post-Pandemic Era

**DOI:** 10.3390/jcm10163466

**Published:** 2021-08-05

**Authors:** Quirino Ciampi, Francesco Antonini-Canterin, Andrea Barbieri, Agata Barchitta, Frank Benedetto, Alberto Cresti, Sofia Miceli, Ines Monte, Licia Petrella, Giuseppe Trocino, Iolanda Aquila, Giovanni Barbati, Valentina Barletta, Daniele Barone, Monica Beraldi, Gianluigi Bergandi, Giuseppe Bilardo, Giuseppe Boriani, Eduardo Bossone, Amedeo Bongarzoni, Francesca Elisa Bovolato, Francesca Bursi, Valeria Cammalleri, Marco Carbonella, Grazia Casavecchia, Sebastiano Cicco, Giovanni Cioffi, Rosangela Cocchia, Paolo Colonna, Lauro Cortigiani, Umberto Cucchini, Maria Grazia D’Alfonso, Antonello D’Andrea, Luca Dell’Angela, Ilaria Dentamaro, Marcella De Paolis, Paola De Stefanis, Wanda Deste, Maria Di Fulvio, Giovanna Di Giannuario, Daniela Di Lisi, Concetta Di Nora, Iacopo Fabiani, Roberta Esposito, Fabio Fazzari, Luigi Ferrara, Gemma Filice, Davide Forno, Mauro Giorgi, Enrico Giustiniano, Cosimo Angelo Greco, Gian Luca Iannuzzi, Annibale Izzo, Alberto Maria Lanzone, Alessandro Malagoli, Francesca Mantovani, Vincenzo Manuppelli, Simona Mega, Elisa Merli, Margherita Ministeri, Doralisa Morrone, Cosimo Napoletano, Luigi Nunziata, Guido Pastorini, Chiara Pedone, Enrica Petruccelli, Maria Vincenza Polito, Vincenzo Polizzi, Costantina Prota, Fausto Rigo, Dante Eduardo Rivaben, Silvio Saponara, Angela Sciacqua, Chiara Sartori, Virginia Scarabeo, Walter Serra, Sergio Severino, Luciano Spinelli, Gloria Tamborini, Antonio Tota, Bruno Villari, Scipione Carerj, Eugenio Picano, Mauro Pepi

**Affiliations:** 1Division of Cardiology, Fatebenefratelli Hospital, Viale Principe di Napoli, 12 I-82100 Benevento, Italy; villari.bruno@gmail.com; 2Rehabilitative Cardiology, Rehabilitative Hospital High Speciality, 31045 Motta di Livenza, Italy; antonini.canterin@gmail.com (F.A.-C.); rivaben1956de@gmail.com (D.E.R.); 3Cardiology Division, University Hospital Modena-Policlinico, 41125 Modena, Italy; olmoberg@libero.it (A.B.); giuseppe.boriani@unimore.it (G.B.); 4Emergency Medicine Division, Azienda Ospedaliera di Padova, 35128 Padova, Italy; agathabarchitta@gmail.com; 5Cardiology Division, Metropolitano Hospital “Bianchi-Melacrino-Morelli”, 89124 Reggio Calabria, Italy; frankbenedetto@gmail.com; 6Cardiology Division, Misericordia Hospital, 58100 Grosseto, Italy; alcresti@gmail.com; 7Geriatric Division, University Hospital Mater Domini, 88100 Catanzaro, Italy; sofy.miceli@libero.it (S.M.); sciacqua@unicz.it (A.S.); 8Cardiology Division, University Hospital Policlinic, University of Catania, 95124 Catania, Italy; inemonte@gmail.com (I.M.); wandadeste@virgilio.it (W.D.); 9Cardiology Division, Civil Hospital G Mazzini, 64100 Teramo, Italy; petrella.licia@gmail.com (L.P.); cosimo.napoletano@aslteramo.it (C.N.); 10Cardiology Division, St Gerardo Hospital, 20900 Monza, Italy; g.trocino@gmail.com; 11Cardiology Division, University Hospital Mater Domini, 88100 Catanzaro, Italy; aquila@unicz.it; 12Cardiology Division, San Bortolo Hospital, 36100 Vicenza, Italy; giovannibarbati@alice.it; 13Cardiology 2 Division, Cardiac Vascular Thoracic Department, Pisa University Hospital, 56124 Pisa, Italy; valentinabarletta@hotmail.com; 14Cardiology Division, S. Andrea Hospital, 19121 La Spezia, Italy; dr.danielebarone@gmail.com; 15Cardiology Division, ASST Mantova, 46100 Mantova, Italy; mberaldi@libero.it; 16Cardiology Division, Civil Hospital, 10015 Ivrea, Italy; gianluigi.bergandi@gmail.com; 17Cardiology Division, Civil Hospital, 32032 Feltre, Italy; bilardogiuseppe@gmail.com; 18Cardiology Division, Cardarelli Hospital, 80131 Napoli, Italy; ebossone@hotmail.com (E.B.); rosangelacocchia@hotmail.com (R.C.); 19Cardiology Division, San Paolo Hospital, ASST San Carlo, San Carlo Unit, 20123 Milano, Italy; amedeo.bongarzoni@tin.it; 20Cardiology Division, Ospedali Riuniti Padova Sud—ULSS Euganea, 35131 Padova, Italy; francescaelisa.bovolato@aulss6.veneto.it; 21Cardiology Division, San Paolo Hospital, ASST Santi Paolo e Carlo, San Paolo Unit, University Center, 20142 Milano, Italy; francescabursi@gmail.com; 22Cardiology Division, University Hospital Policlinico Tor Vergata, 00133 Rome, Italy; v.cammalleri@hotmail.com; 23Cardiology Division, SS Maria Addolorata Hospital, 84025 Eboli, Italy; marcocarbonella210873@gmail.com; 24Cardiology Division, Riuniti Hospital, 71122 Foggia, Italy; graziacasavecchia@libero.it; 25Internal Medicine Division, Policlinic Hospital, 70124 Bari, Italy; sebacicco@gmail.com; 26Cardiology Division, San Pancrazio Arco di Trento Hospital, 38062 Arco di Trento, Italy; dottcioffi@gmail.com; 27Cardiology Division, Policlinic Hospital, 70124 Bari, Italy; colonna@tiscali.it (P.C.); totaantonio@libero.it (A.T.); 28Cardiology Division, San Luca Hospital, 55100 Lucca, Italy; lacortig@tin.it; 29Cardiology Division, San Bassiano Hospital, 36061 Bassano Del Grappa, Italy; ucucchini@yahoo.it; 30Non-Invasive Cardiovascular Diagnostic Division, Careggi Hospital, 50139 Firenze, Italy; mariagrazia.dalfonso@gmail.com; 31Cardiology Division, Umberto I Hospital, 84014 Nocera Inferiore, Italy; antonellodandrea@libero.it; 32Cardiology Division, Gorizia-Monfalcone Hospital, 34170 Gorizia, Italy; lucadellangela@libero.it; 33Cardiology Division, Miulli Hospital, 70021 Acquaviva delle Fonti, Italy; ilaria.dentamaro@hotmail.it; 34Cardiology Division, Santa Maria Hospital, 05100 Terni, Italy; marcelladepaolis@gmail.com; 35Cardiology Division, San Luigi Gonzaga University Hospital, 10043 Orbassano, Italy; paoladestefanis@libero.it; 36Cardiology Division, Clinicizzato Hospital, 66100 Chieti, Italy; mariadifulvio@gmail.com; 37Cardiology Division, Infermi Hospital, 47923 Rimini, Italy; gdigiannuario@gmail.com; 38Cardiology Division, Policlinico Hospital, 90127 Palermo, Italy; danydilis@hotmail.it; 39Cardiology Division, University Hospital, 33100 Udine, Italy; concetta.dinora@gmail.com; 40Cardiology and Cardiovascular Medicine Division, Fondazione Gabriele Monasterio, 56124 Pisa, Italy; iacopofabiani@gmail.com; 41Internal Medicine Division, Federico II University Hospital, 80138 Napoli, Italy; roberta.esposito6@gmail.com; 42Echocardiography Division, Humanitas Clinical and Research Center, 20089 Rozzano, Italy; fabiofazzari@hotmail.it; 43Cardiology Division, Villa Dei Fiori Clinic, 80011 Acerra, Italy; luigiferraradoc@alice.it; 44Cardiology Division, Annunziata Hospital, 87100 Cosenza, Italy; gemma@gmx.it; 45Cardiology Division, Maria Vittoria Hospital, 10144 Torino, Italy; fornodavide@yahoo.com; 46Cardiology Division, Città della Scienza e della Salute Hospital, 10126 Torino, Italy; mauro.giorgi@yahoo.it; 47Department of Anesthesiology, Humanitas Clinical and Research Center, 20089 Rozzano, Italy; enrico.giustiniano@gmail.com; 48Cardiac Surgery Division, Vito Fazzi Hospital, 73100 Lecce, Italy; cosimoangelo.greco@gmail.com; 49Cardiology Division, Maugeri Clinic, 82037 Telese, Italy; gianluca.iannuzzi@icsmaugeri.it; 50Cardiology Division, Sant’Anna e San Sebastiano Hospital, 81100 Caserta, Italy; anniizzo@libero.it; 51Cardiology Division, Humanitas Gavazzeni Hospital, 24125 Bergamo, Italy; albertomaria.lanzone@alice.it; 52Cardiology Division, S. Agostino Estense Hospital, 41126 Baggiovara, Italy; ale.malagoli@gmail.com; 53Cardiology Division, Azienda USL—IRCCS di Reggio Emilia, 42122 Reggio Emilia, Italy; francy_manto@hotmail.com; 54Cardiology Division, University Hospital Ospedali Riuniti di Foggia, 71122 Foggia, Italy; manuppelli.dr@gmail.com; 55Cardiology Division, Campus Biomedico University Hospital, 00128 Roma, Italy; s.mega@unicampus.it; 56Cardiology Division, Degli Infermi Hospital, 48018 Faenza, Italy; elisamerli@libero.it; 57Cardiology Division, Centro Cuore Morgagni, 95030 Pedara, Italy; margheritaministeri@yahoo.it; 58Cardiology Division, Department of Surgical, Medical and Molecular Pathology and Critical Care Medicine, University of Pisa, 56126 Pisa, Italy; doralisamorrone@gmail.com; 59Cardiology Division, Santa Maria della Pietà Hospital, 80035 Nola, Italy; nunziata.luigi@alice.it; 60Cardiology Division, Regina Montis Regalis Hospital, 12084 Mondovì, Italy; guido.pastorini@gmail.com; 61Cardiology Division, Maggiore Hospital, 40133 Bologna, Italy; chiara.pedone@ausl.bologna.it; 62Cardiology Division, S. Giacomo Hospital, 70043 Monopoli, Italy; epetruccelli@libero.it; 63Cardiology Division, Evangelic Hospital Betania, 80147 Napoli, Italy; mvpolito@hotmail.it; 64Cardiology Division, San Camillo-Fornalinini Hospital, 00152 Roma, Italy; vincenzopolizzi@libero.it; 65Cardiology Division, San Luca Hospital, 84078 Vallo della Lucania, Italy; costantinaprota@gmail.com; 66Cardiology Division, Civil Hospital Dolo, 30031 Dolo, Italy; fausto.rigo@aulss3.veneto.it; 67Cardiology Division, Luigi Curto Hospital, 84035 Polla, Italy; cardiopolla@gmail.com; 68Cardiology Division, Santi Antonio e Biagio e Cesare Arrigo Hospital, 15121 Alessandria, Italy; chiara.sartori@ospedale.al.it; 69Cardiology Division, Camposampiero Hospital, 35012 Camposampiero, Italy; virginia.scarabeo@libero.it; 70Cardiology Division, University Hospital Parma, 43126 Parma, Italy; wserra@ao.pr.it; 71Cardiology Division, Cotugno Hospital, 80128 Naples, Italy; srgsev@yahoo.it; 72Rehabilitative Cardiology, S. Rita Clinic, 88811 Cirò Marina, Italy; luciano.spinelli@virgilio.it; 73Cardiology Division, Centro Cardiologico Monzino, Istituto di Ricovero e Cura a Carattere Scientifico (IRCCS), 20138 Milano, Italy; gloria.tamborini@ccfm.it (G.T.); mauro.pepi@ccfm.it (M.P.); 74Department of Clinic and Experimental Medicine, University of Messina, 98122 Messina, Italy; scipione2@interfree.it; 75Biomedicine Department, National Research Council Institute of Clinical Physiology (CNR-IFC), 56124 Pisa, Italy; picano@ifc.cnr.it

**Keywords:** COVID-19, point-of-care cardiac ultrasound, lung ultrasound

## Abstract

Background: Cardiology divisions reshaped their activities during the coronavirus disease 2019 (COVID-19) pandemic. This study aimed to analyze the organization of echocardiographic laboratories and echocardiography practice during the second wave of the COVID-19 pandemic in Italy, and the expectations for the post-COVID era. Methods: We analyzed two different time periods: the month of November during the second wave of the COVID-19 pandemic (2020) and the identical month during 2019 (November 2019). Results: During the second wave of the COVID-19 pandemic, the hospital activity was partially reduced in 42 (60%) and wholly interrupted in 3 (4%) echocardiographic laboratories, whereas outpatient echocardiographic activity was partially reduced in 41 (59%) and completely interrupted in 7 (10%) laboratories. We observed an important change in the organization of activities in the echocardiography laboratory which reduced the operator-risk and improved self-protection of operators by using appropriate personal protection equipment. Operators wore FFP2 in 58 centers (83%) during trans-thoracic echocardiography (TTE), in 65 centers (93%) during transesophageal echocardiography (TEE) and 63 centers (90%) during stress echocardiography. The second wave caused a significant reduction in number of echocardiographic exams, compared to November 2019 (from 513 ± 539 to 341 ± 299 exams per center, −34%, *p* < 0.001). On average, there was a significant increase in the outpatient waiting list for elective echocardiographic exams (from 32.0 ± 28.1 to 45.5 ± 44.9 days, +41%, *p* < 0.001), with a reduction of in-hospital waiting list (2.9 ± 2.4 to 2.4 ± 2.0 days, −17%, *p* < 0.001). We observed a large diffusion of point-of-care cardiac ultrasound (88%), with a significant increase of lung ultrasound usage in 30 centers (43%) during 2019, extended to all centers in 2020. Carbon dioxide production by examination is an indicator of the environmental impact of technology (100-fold less with echocardiography compared to other cardiac imaging techniques). It was ignored in 2019 by 100% of centers, and currently it is considered potentially crucial for decision-making in cardiac imaging by 65 centers (93%). Conclusions: In one year, major changes occurred in echocardiography practice and culture. The examination structure changed with extensive usage of point-of-care cardiac ultrasound and with lung ultrasound embedded by default in the TTE examination, as well as the COVID-19 testing.

## 1. Introduction

Cardiologists were engaged in an important way in the coronavirus disease 2019 (COVID-19) pandemic, with an important tribute in terms of infected and passed away professionals involved in clinical management and diagnostic screening with cardiac and lung ultrasound. During the first lockdown period (March/April 2020) in Italy, as previously reported worldwide, there was an unexpected drop in emergency cardiology admissions [1,2]. The reshaping of cardiology activities caused an important impact on echocardiographic laboratories [3]. In fact, in a previous survey, we observed a significant reduction in every echocardiographic exam during the lockdown period [4].

A new intriguing hypothesis was proposed about the cardio-protective benefits of lockdown, which induced air cleaning from pollution [5]. However, the awareness on the potential environmental impact of medical imaging is still poorly defined. Therefore, we sought to provide an instant survey of the echocardiography practice during the second wave of the COVID-19 pandemic in Italy, with high expectations for the post-COVID era regarding the echocardiographic activity and the perceived role of carbon dioxide production by cardiac imaging modalities in decision-making and its importance in the future development of imaging.

This study aimed to analyze the organization of echocardiographic laboratories and echocardiography practice during the second wave of the COVID-19 pandemic in Italy, and the expectations for the post-COVID era.

## 2. Methods

We analyzed two different time periods: the month of November during the second wave of the COVID-19 pandemic (2020) and the identical month during 2019 (November 2019).

A list of accredited echocardiographic laboratories was obtained from the Italian Society of Echocardiography and Cardiovascular Imaging (SIECVI). Each member of SIECVI was contacted by mail.

Data were retrieved via an electronic survey based on a structured questionnaire that was uploaded on the SIECVI website (www.siec.it, accessed on 14 December 2020)

For allocation of the response, the questionnaire required general information, such as the name of the hospital, the investigator and the interviewed person’s name:General information: date, hospital’s name, department, name of the interviewed physician, and city and region of Italy.Hospital activity and outpatient echocardiographic activity during the second wave of the COVID-19 pandemic.The number of echocardiographic exams and the duration of waiting in-hospital lists and for outpatients in the two analyzed periods.Types of activities organization in the echocardiography laboratory to reduce the operator risk: social distancing in the waiting room, limit to accompanying visitors, wearing of masks, reducing the number of exams, improvement of operators’ self-protection and nasopharyngeal swab required for patients before echocardiographic exams.Usage of point-of-care cardiac ultrasound by cardiologists with a joint reading assessment with other physicians.Use of lung ultrasound.Modality of analysis of echocardiographic imaging.Expectations for the post-COVID era regarding the echocardiographic activity.Role of carbon dioxide production by cardiac imaging modalities in decision-making and its importance in the future development of imaging.

### Statistical Analysis

Data were expressed as a mean ± standard deviations for continuous variables and as numbers (percentage) for categorical variables. Continuous variables were compared by using the Student’s unpaired test, while differences of categorical variables were assessed by the chi-square test.

A probability value of <0.05 was considered statistically significant.

All statistical calculations were performed by using SPSS for Windows, release 18.0 (Chicago, IL, USA).

## 3. Results

Data were obtained from 70 echocardiographic laboratories: 39 centers (56%) were in the northern regions of Italy (Lombardy, Veneto, Piedmont, Liguria, Friuli-Venezia-Giulia, Trentino-Alto-Adige and Emilia Romagna), 11 centers (16%) were in the central regions (Abruzzo, Latium, Tuscany and Umbria) and 21 (30%) were in southern regions (Campania, Sicily, Apulia and Calabria).

All centers had COVID-divisions, and six centers (9%) were COVID-dedicated hospitals, with three centers in the northern regions and three in the central and southern regions.

The cardiologists of the involved centers in the survey showed a relevant impact of COVID disease (138/991, 13.9%). Moreover, the cardiologists dedicated to the echocardiographic laboratories presented the same incidence of SARS-COVID disease, compared to other cardiologists (48/284, 16.9% vs. 90/707, 12.7%, *p* = 0.264).

During the second wave of the COVID-19 pandemic, the hospital activity proceeded normally in 25 centers (36%), partially reduced in 42 (60%) and wholly interrupted in 3 (4%), whereas outpatient echocardiographic activity was normal in 22 centers (31%), partially reduced in 41 (59%) and completely interrupted in 7 (10%).

The effect of the second wave caused a significant reduction in the echocardiographic exams, compared to November 2019 (from 513 ± 539 to 341 ± 299 exams per center, −34%, *p* < 0.001, Figure 1). On average, there was a significant increase in outpatient waiting lists for elective echocardiographic exams (from 32.0 ± 28.1 to 45.5 ± 44.9 days, +41%, *p* < 0.001, Figure 1), with reduction of in-hospital waiting lists (2.9 ± 2.4 to 2.4 ± 2.0 days, −17%, *p* < 0.001, Figure 1).

We did not find a difference in the centers with COVID-dedicated hospitals compared to centers without being involved in the number of echocardiographic exams (350 ± 228 vs. 341 ± 286 exams, *p* = 0.946) in outpatient waiting list for elective echocardiographic exams (75.8 ± 52.3 vs. 40.7 ± 42.0 days, *p* = 0.100) and in-hospital waiting list (2.8 ± 1.9 to 2.4 ± 2.0 days, *p* = 0.214).

We observed an important change in the activities’ organization in the echocardiography laboratory to reduce the operator risk: social distancing in waiting rooms (65 centers, 93%), limit to accompanying visitors (69, 99%), wearing of masks (all centers, 100%) and reducing the number of exams (59, 84%). improvement of self-protection of operators using appropriate personal protection equipment: FFP2 in 58 centers (83%) during trans-thoracic echocardiography (TTE), 65 (93%) during transesophageal echocardiography (TEE) and 63 (90%) during stress echocardiography (SE).

A nasopharyngeal swab was required for patients before echocardiographic exams in 46 centers (66%): only before TEE in 25 centers (54%), before TEE and SE in 14 centers (31%) and before all echocardiographic exams in 7 centers (15%). A nasopharyngeal swab was organized directly by the hospital into 46 centers (67%). On balance of test safety with specific reference to contagion risk, 87% of centers considered TTE safer than TEE or SE.

About SE, pharmacological SE was considered safer than exercise in 57 centers (81%).

We observed a large and significant diffusion of point-of-care cardiac ultrasound (88%). It was performed by cardiologists with joint reading assessment with anesthesiologists (41%) or intensivists (36%) (Figure 2). Imaging analysis was mainly online (42 centers, 59%), but also offline (26, 38%) or by teleconsulting (2, 3%).

In 2019, the usage of lung ultrasound was a routine in 30 centers (43%); in 2020, all centers integrated lung ultrasound in the standard TTE evaluation of every patient, with or without suspicion of COVID-19 infection (Figure 2).

We also analyzed the expectations for the post-COVID era: participant centers felt that there should be a stable reduction of inappropriate echocardiographic exams (16 centers, 23%), no change in activity in the next year (23, 33%) or project a rebound of activity to take care of the follow-up examinations skipped during the pandemic (30, 43%).

Moreover, 65 centers (93%) believe that diagnostic imaging significantly contributes to the production of carbon dioxide, which is an indicator of the environmental impact of technology, and it should be incorporated in an integrated decision-making between alternative cardiac imaging modalities (Figure 2).

## 4. Discussion

In this manuscript, we demonstrated that echocardiographic laboratories in Italy were readily highly prepared for the second wave of the COVID-19 pandemic. In fact, we encountered an important change in the organization of echocardiographic activities to reduce patients’ risks: reduction of the exams’ number, social distancing in waiting rooms, limitation of accompanying visitors, wearing of masks and prescribing nasopharyngeal swab. Operators improved self-protection with the systematic de facto obligatory usage of appropriate personal protection equipment during echocardiographic exams.

This new organization of echocardiographic laboratories agreed with the previous papers [6,7] and the position papers of SIECVI [3] that identified these measures as the most important elements to reduce COVID-related risks. Our survey is focused on the Italian Echocardiography community, but similar precautions, recommendations and protection policies were encouraged and applied by single institutions [8], European Association of Echocardiography and Cardiovascular Imaging [9] and American Society of Echocardiography [7].

One more important point was the reduction of the exams’ number: deferring echocardiography studies deemed elective and non-urgent (or inappropriate), greatly reducing volumes in an effort to protect patients and echocardiography laboratory staff [10].

There was a profound remodeling also in the use of stress modalities in the Echocardiography laboratories, since exercise is considered a high-risk procedure because of aerosolization, and pharmacological stress became, by far, the test of choice in most laboratories [6].

The COVID pandemic has paradoxically produced an opportunity to help improve both sustainability and equity in the healthcare field and to reduce unnecessary and/or inappropriate tests, and deferring elective procedures [11].

TTE plays a prominent role in identifying cardiac complications related to COVID-19, but during TTE, the distance physician–patient is so reduced, getting in the droplets’ area. In addition to the preventive measures listed above, cardiac point-of-care ultrasound offers irreplaceable benefits to reduce the duration of TTE and, consequently, the time of exposure. Cardiac point-of-care ultrasound allows for rapid focused diagnostic assessment by providers already at the bedside [12,13]. Accordingly, in this survey, we observed a large diffusion of point-of-care cardiac ultrasound (88% of the centers) with the possibility to perform the examination not only by cardiologists but also with joint reading assessments with anesthesiologists or intensivists.

We have taken notice of a significant increase in the usage of lung ultrasound associated to TTE in all patients during COVID’s second wave (+57%) compared to the same period in 2019. Lung ultrasound is a useful diagnostic tool in different clinical conditions [14], as well as in the COVID-19 pandemic: in the first triage of symptomatic patients, in the emergency department, in the prognostic stratification and monitoring of patients with pneumonia, and in the management of patients in the intensive care unit, with low-cost and radiation-free approach. Another additional application of lung ultrasound is in pre-hospital diagnosis and, also, home monitoring of COVID patients [15,16]

Moreover, “bedside” lung ultrasound can reduce the number of physicians exposed to the virus during patients’ assessment and treatment.

The increase in the usage of lung ultrasound should be considered as the common language and meeting point between cardiologists and intensivist non-cardiology physicians aware of the importance and clinical role of lung ultrasound in critical healthcare [14].

A large portion of the centers (95%) believes that cardiological imaging played a role into the production of carbon dioxide. Indeed, the healthcare industry contributes 5 to 10% of the global carbon dioxide emission, and medical imaging for about 1%. Carbon dioxide emissions are the primary factor for global climate change. It is widely recognized that reducing carbon dioxide emissions is important to attenuate the impacts of climate change. The worsening of air quality induced by pollution acutely increases the admission rates for acute coronary syndromes, acute decompensated heart failure and atrial fibrillation [5]. Conversely, the improvement of air quality reduces the admission rates for the same conditions, as proven during the COVID-19 pandemic, due to the tumble of air pollution due to the lockdown [17,18]. Changes in hospital admission are only the tip of the iceberg of cardiovascular toxic effects of pollution, since, in patients with chronic conditions, coronary artery diseases or heart failure may show an increased vulnerability in coronary flow reserve or pulmonary congestion in the presence of increased air pollution levels [19]. Up until now, prescriptions in cardiac imaging have completely neglected the environmental dimension, yet we know that the environmental impact expressed as the emission of CO_2_ equivalents varies by a factor of 100 or 1000 between different cardiac imaging techniques, with an echocardiographic examination associated with 2 kg of carbon dioxide emissions and a cardiac resonance imaging or cardiac computed tomography associated with at least 100-fold higher level of emissions [20,21]. The variable of environmental impact of cardiac imaging is now increasingly perceived as being significant by the Italian echocardiographic community. There is a diffuse sentiment that scientific societies may play a major role in disseminating the culture of environmental sustainability in cardiac imaging, eventually promoting climate-neutral choices from industries, doctors and patients also in this important sector of economy [22].

## 5. Conclusions

In one year, major changes occurred in echocardiography practice and culture, triggered by the adaptive response to the COVID-19 pandemic. Fewer exams are performed per day, with a significant increase of outpatient waiting list for elective echocardiographic exams. Echocardiographic laboratories in Italy minimized the contagion risk by adequate protection policies, and the risk of infection is now included by prescribers and practitioners in the risk–benefit assessment of testing.

The perspective of the study offers a selection of important echocardiographic exams, a better and stricter adherence to the criteria of appropriateness, building the application of a flowchart for the selection of important exams and contributing to the setting aside of clearly inappropriate examinations.

The perceived risk of contagion is higher with TEE than with TTE. The exams structure changed, and lung ultrasounds for B-lines and pleural effusions are now embedded by default in the TTE examination also outside the COVID-19 testing. It is now an almost unanimous sentiment that the environmental impact of cardiac imaging should be incorporated into the risk–benefit assessment of cardiac imaging, with preferential steering towards climate-neutral choices. Carbon dioxide production by cardiac imaging examination is important to be aware of and consider its effect in environmental impact assessment in healthcare, and it should be disseminated by scientific imaging societies.

## Figures and Tables

**Figure 1 jcm-10-03466-f001:**
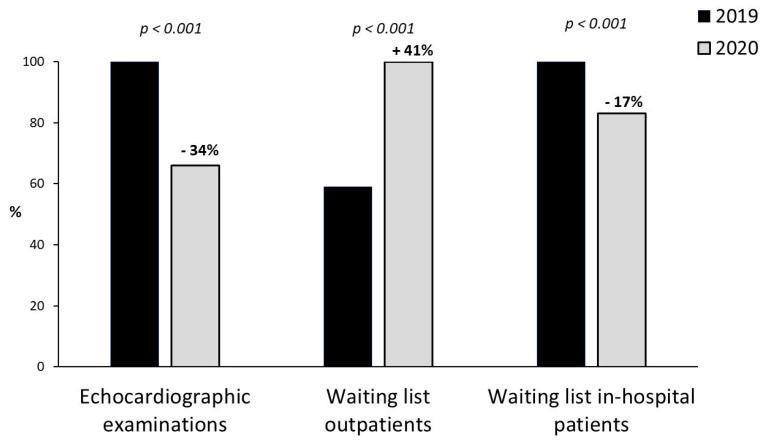
Percentage difference from November 2019 (black bar) to November 2020 (gray bars) in echocardiographic examinations, waiting list outpatients and waiting list in-hospital patients.

**Figure 2 jcm-10-03466-f002:**
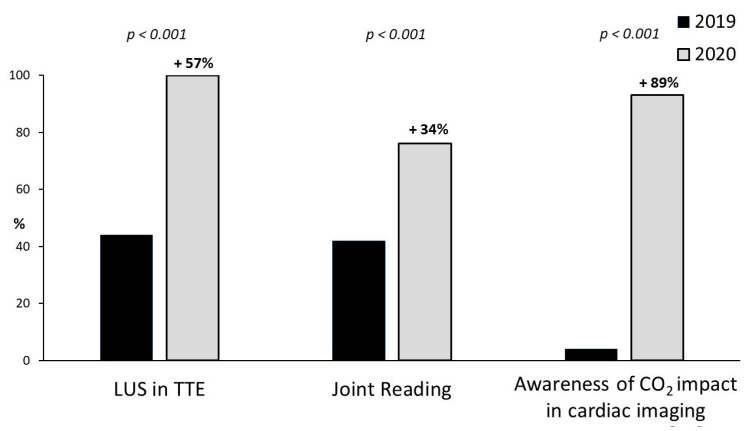
Percentage difference from November 2019 (black bar) to November 2020 (gray bars) in lung ultrasound evaluation in resting TTE, joint reading assessment with intensivists and awareness of carbon dioxide (CO_2_) impact in cardiac imaging.

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
