# Peer review of "Reshaping of Italian Echocardiographic Laboratories Activities during the Second Wave of COVID-19 Pandemic and Expectations for the Post-Pandemic Era"

_jcm, 2021, doi:10.3390/jcm10163466_

Round 1
Reviewer 1 Report
The authors describe how the Covid-19 pandemic has impacted the work flow in 70 Italian echocardiography centers during the second wave and how they expect that it will change in the future. The subject is interesting and the considerations regarding environmental impact are very original.
I have following suggestions:
- several grammatical errors throughout the text, I would recommend revision by a native english speaker.
-
In the results, the authors report that a nasopharyngeal swab was required for patients before echocardiographic exams in 46 centers (66%). It is however unclear whether this applies to both transthoracic and transesophageal echocardiograms. I would expect that almost all centers require this for TOE, but not for TTE. Please clarify.
- what do the authors mean by 'sfigle' line 220?
-
The authors report that almost all centers believe that diagnostic imaging significantly contributes to the production of carbon dioxide and should be incorporated in an integrated decision-making between alternative cardiac imaging modalities. This remains rather vague, and I would rather describe this as 'awareness' of CO2 impact of cardiac imaging, rather than 'knowledge' unless the survey really tested the knowledge regarding CO2 emissions for the various cardiac imaging modalities.
- this study solely applies to the situation in Italy, but it would also be interesting if the authors would interpret their findings in the broader international context. Are there any data from other countries regarding organisation of the work flow in the covid era?
Author Response
Reviewer #1:
Thank you for your thoughtful and constructive criticism. The parts of your commentary are reported in bold. All the changes in the revised manuscript are reported in red font.
Several grammatical errors throughout the text, I would recommend revision by a native English speaker.
I appreciate that you underlined this aspect. The manuscript was revised by a native English speaker.
In the results, the authors report that a nasopharyngeal swab was required for patients before echocardiographic exams in 46 centres (66%). It is, however, unclear whether this applies to both transthoracic and transesophageal echocardiograms. I would expect that almost all centres require this for TOE, but not for TTE. Please clarify.
We changed in the Results’ section according your suggestion: …” A nasopharyngeal swab was required for patients before echocardiographic exams in 46 centers (66%): only before ETE in 25 centers (54%), before ETE and SE in 14 centers (31%) and before all echocardiographic exams in 7 centers (15%).”….
What do the authors mean by 'sfigle' line 220?
We deleted 'sfigle'. It is a typing error
The authors report that almost all centres believe that diagnostic imaging significantly contributes to the production of carbon dioxide, and it should be incorporated in an integrated decision-making among alternative cardiac imaging modalities. This remains rather vague, and I would rather describe this as an 'awareness' of CO2 impact of cardiac imaging, rather than 'knowledge' unless the survey really tested the knowledge regarding CO2 emissions for the various cardiac imaging modalities.
We acknowledged this and are aware of the points outlined.
This study solely applies to the situation concerning Italy, but it would also be interesting if the authors would interpret their findings in the broader international context. Are there any data from other countries regarding organization of the work flow in the COVID era?
We changed in the Discussion section: “Our survey is focused on the Italian Echocardiography community, but similar precautions, recommendations and protection policies were encouraged and applied by single institutions (8), European Association of Echocardiography and Cardiovascular Imaging (9) and American Society of Echocardiography (7)”….

Reviewer 2 Report
In this nice paper, the authors deals with the effect of the Covid-19 pandemic on the organization and activity of Echo Labs in Italy.
The paper is interesting because it provide a picture of the evolution of echo practice during the COvid 19 pandemic.
- The authors should indicate which Region were involved in the study in the South and Center of Italy.
- Are there any difference in the Echo Lab organization between Covid-dedicated and Covid non-dedicated centers? For instance, did the reduction of activity and the delayed outpatient waiting impact more significantly Covid dedicated centers?
- Page 4, line 190: “The effect of the second wave caused a significant reduction IN THE number of echocardiographic exams”
- Page 4, line 191 “Figfigures 1 »
- During the second pandemic wave, there was a reduction in the number of exams. What kind of exams where keeped? Can the authors give a list of the most applied criteria to select the exam to keep?
- The authors state that FFP2 masks were used in 90% of the centers during stress echocardiography (SE). What about exercise echocardiography?
- Exercise echo is theoretically more contaminating than stress echo. Did the reduction of activity affect more exercise echo that stress echo?
- It seems that Figure 1 and 2 report the same information. The authors should delate one of these figures.
- Did the authors have any data on the number of echo-operator (cardiologist/sonographer) developing SARS-COVID disease during the pandemic? It will be interesting to have this data and compare it with the mean rate of infection in cardiology and in the general medial population.
- Which are the perspective of the study? From a practical point of view, will the centers maintain the criteria applied for the selection of important echo exams in the future? For instance through the application of a flow chart for the selection of important exams?
Author Response
Reviewer #2:
Thank you for your thoughtful and constructive criticism. In bold your comments. All the changes in the revised manuscript are reported red font.
The authors should indicate which Region were involved in the study in the South and Center of Italy.
We indicated all Regions of Italy involved in the northern, center and southern. “…Data were obtained from 70 echocardiographic laboratories: 39 centers (56%) were in northern regions of Italy (Lombardy, Veneto, Piedmont, Liguria, Friuli-Venezia-Giulia, Trentino- Alto-Adice, and Emilia Romagna), 11 centers (16%) were in central regions (Abruzzo, Latium, Tuscany, Umbria) and 21 (30%) in southern regions (Campania, Sicily, Apulia, and Calabria)…”
Are there any difference in the Echo Lab organization between Covid-dedicated and Covid non-dedicated centers? For instance, did the reduction of activity and the delayed outpatient waiting impact more significantly Covid dedicated centers?
We analyzed this point. In the Results section we added: ..“We did not find difference in the centers with COVID-dedicated hospitals compared to centers without in the number of echocardiographic exams (350±228 vs 341±286 exams, p=0.946) in outpatient waiting list for elective echocardiographic exams (75.8±52.3 vs 40.7±42.0 days, p=0.100), and in-hospital waiting list (2.8±1.9 to 2.4±2.0 days, p=0.214).”
Page 4, line 190: “The effect of the second wave caused a significant reduction IN THE number of echocardiographic exams”
We changed in the text according to your suggestions
Page 4, line 191 “Figfigures 1 »
We changed FigFigures with Figures.
During the second pandemic wave, there was a reduction in the number of exams. What kind of exams where keeped? Can the authors give a list of the most applied criteria to select the exam to keep?
We did not have this data. We asked the expectations for the post-COVID era about application of appropriateness criteria, volume of activity and project a rebound of activity in the post-COVID.
The authors state that FFP2 masks were used in 90% of the centers during stress echocardiography (SE). What about exercise echocardiography?
Exercise echo is theoretically more contaminating than stress echo. Did the reduction of activity affect more exercise echo that stress echo?
We did not have quantitative numbers, but we could add a sentence in the Discussion section as follows:
.. “There was a profound remodeling also in the use of stress modalities in the Echocardiography laboratories, since exercise is considered a high-risk procedure because of aerosolization and pharmacological stress became by far the test of choice in most laboratories (6).”…..
It seems that Figure 1 and 2 report the same information. The authors should delate one of these figures.
Thanks for your suggestion. We eliminated the figure 1.
Did the authors have any data on the number of echo-operator (cardiologist/sonographer) developing SARS-COVID disease during the pandemic? It will be interesting to have this data and compare it with the mean rate of infection in cardiology and in the general medial population.
We asked all the centers involved in the survey to have this additional data data by email. We received data from 62/70 centers (89%). We added in the Results sections: ..“The cardiologists of the involved centers showed a relevant impact of SARS-COVID disease (138/991, 13.9%). In addition, cardiologists dedicated to the echocardiographic laboratories presented the same incidence of SARS-COVID disease, compared to other cardiologists (48/284, 16.9% vs 90/707, 12.7%, p=0.264)”….
Which are the perspective of the study? From a practical point of view, will the centers maintain the criteria applied for the selection of important echo exams in the future? For instance through the application of a flow chart for the selection of important exams?
In the conclusion session we added: …. “The perspective of the study will be a selection of important echocardiographic exams, a better and stricter adherence to the criteria of appropriateness, building the application of a flow chart for the selection of important exams, contributing to the setting aside of clearly inappropriate examinations.”…

Round 2
Reviewer 2 Report
The authors have answered all the questions raised by the reviewer.
I have only a remark left : page 5, line 214, ETE should stand for TEE, I guess. Please amend.